# Crack Detection and Comparison Study Based on Faster R-CNN and Mask R-CNN

**DOI:** 10.3390/s22031215

**Published:** 2022-02-05

**Authors:** Xiangyang Xu, Mian Zhao, Peixin Shi, Ruiqi Ren, Xuhui He, Xiaojun Wei, Hao Yang

**Affiliations:** 1School of Rail Transportation, Soochow University, Suzhou 215006, China; X.Y.Xu@suda.edu.cn (X.X.); 20215246020@stu.suda.edu.cn (M.Z.); pxshi@suda.edu.cn (P.S.); 20204046002@stu.suda.edu.cn (R.R.); 2School of Civil Engineering & Transportation, Central South University, Changsha 410075, China; xuhuihe@csu.edu.cn (X.H.); xiaojun.wei@csu.edu.cn (X.W.); 3School of Transportation and Civil Engineering, Nantong University, Nantong 226019, China

**Keywords:** deep learning, Mask R-CNN, crack detection, Faster R-CNN, intelligent monitoring

## Abstract

The intelligent crack detection method is an important guarantee for the realization of intelligent operation and maintenance, and it is of great significance to traffic safety. In recent years, the recognition of road pavement cracks based on computer vision has attracted increasing attention. With the technological breakthroughs of general deep learning algorithms in recent years, detection algorithms based on deep learning and convolutional neural networks have achieved better results in the field of crack recognition. In this paper, deep learning is investigated to intelligently detect road cracks, and Faster R-CNN and Mask R-CNN are compared and analyzed. The results show that the joint training strategy is very effective, and we are able to ensure that both Faster R-CNN and Mask R-CNN complete the crack detection task when trained with only 130+ images and can outperform YOLOv3. However, the joint training strategy causes a degradation in the effectiveness of the bounding box detected by Mask R-CNN.

## 1. Introduction

In the research field of road crack recognition that is primarily based on computer vision, the essential methods mainly include digital image processing methods, which mainly distinguish features manually and employ many feature rules to design some feature recognition conditions, and convolutional networks based on deep learning, which adopt networks to automatically investigate the features of information so that the net can continuously adjust itself according to a certain rule to achieve input data output equal to or close to the label. In the previous ten years, many scholars have conducted in-depth examinations of road crack recognition primarily based on digital image processing. Hoang [1] proposed a smart method of automatically classifying road cracks to enhance the effectiveness of periodic surveys of asphalt pavement conditions. The new method depends on algorithms of computational intelligence and methods of image processing. Lei et al. [2] proposed a new crack detection approach based completely on the crack central point—particularly, the crack central point approach—to address these critical issues. With a small quantity of images, the new approach can rapidly and precisely pick out cracks in the gathered images. Furthermore, Ying et al. [3] introduced a novel way to detect and classify cracks by digital images, which makes use of an image magnification algorithm. However, with the rapid growth of road mileage, the information on road pavement cracks is already massive, so traditional coding ideas based on experience to set logic have difficulty meeting actual needs. At the same time, computer hardware is also developing rapidly, leading to breakthroughs in deep learning-related algorithms. In recent years, widely applicable deep learning algorithms have become increasingly popular, and computer hardware and convolutional neural networks have become the core of deep learning [4]. Therefore, detection algorithms based on deep learning and convolutional neural networks have achieved better results in the field of crack recognition.

To quantify the defect width or length, many researchers have proposed deep learning-based quantification methods. Kim et al. [5] introduced a novel crack evaluation framework for concrete constructions that detects cracks using masks and location-based totally convolutional neural communities (Mask R-CNN) and quantifies cracks using a few morphological operations on the detected crack masks. Kalfarisi et al. [6] proposed a unified framework for the usage of actuality mesh-modeling technological know-how that enables quantitative evaluation with the integrated visualization of an inspected structure. The effectiveness and robustness of the developed methods are evaluated, and the usage of various actual instances, including road pavements, bridges, underground tunnels, buildings, and water towers, is established. Wu et al. [7] adopted images to assemble a mesh model through a photogrammetry approach. The model with annotated cracks allows intuitive visualization and quantitative evaluation of thousands of detected cracks. Moreover, Peng et al. [8] proposed a computer vision approach for bridge crack cognizance and width quantification via hybrid characteristic learning. Furthermore, Guo et al. [9] introduced a computer vision approach for the identification, quantification, and visualization of microcracks, primarily based on deep learning. Recently, Bang et al. [10] proposed deep learning-based detection and the quantification of structural damage by adopting structured lights and a depth camera.

Many other scholars have improved the performance of the model with better detection results. Han et al. [11] proposed a sampling block with the implementation of convolutional neural networks, thereby developing a novel pixel-level semantic segmentation network. Li et al. [12] proposed a multilayer characteristic fusion network based totally on the faster region-based convolutional neural network (Faster R-CNN) to gain automated tunnel surface defect detection with excessive precision. Ju Huyan et al. [13] proposed the fusion of a sensitivity detection network, which is capable of monitoring unsealed and sealed cracks with severely complicated backgrounds. Moreover, Malini et al. [14] used a regularization approach to obtain greater total performance primarily based on a convolutional neural network (CNN) monitoring model. Cha et al. [15] proposed a vision-based approach using a deep CNN architecture for monitoring concrete cracks except for calculating the defect features. Mogalapalli et al. [16] proposed a quantum transfer learning-based method for various image classification tasks. Pang et al. [17] proposed a two-stage crack defect segmentation approach based totally on the target detection community to resolve the issue of extreme brightness imbalance and massive noise in dam surface images. Sekar et al. [18] introduced a new multitasking Faster R-CNN method using the region of interest and global average pooling align methods to monitor various road cracks.

Many researchers have added classification functions for extraordinary defects. Trana et al. [19] proposed a two-step sequential computerized method for detecting cracks and classifying the severity of asphalt pavements. Liu et al. [20] introduced a new model combining pixelwise and region-based deep learning to supply pavement inspection technological know-how for mutually acquiring misery classes, locations, and geometric information. Moreover, Mondal et al. [21] analyzed visual data captured by means of sensors established on robots, and the damages could be detected and categorized autonomously. The current study proposes the use of deep learning-based methods to this end. Hou et al. [22] introduced transfer learning with the Cascade Mask RCNN mannequin for defect identification and location. Dhiman et al. [23] proposed two methods primarily based on stereo-vision evaluation of street environments beforehand of the vehicle, and they additionally designed two models for deep-learning-based pothole detection. Intelligent monitoring methods [24,25,26] will be an essential guarantee for the realization of smart operation and maintenance.

Faster R-CNN and Mask R-CNN have been proven successful in the field of general vision, such as in [27]. However, the crack detection research considering the characteristics of road images is still insufficient. We experimentally validated our model using these two methods for the specific task of road crack detection. To speed up the training, we trained our model by a joint training strategy using pre-training and migration learning. The results show that we were able to make Faster R-CNN and Mask R-CNN with good results using only 130+ images.

## 2. Method

The pipeline of Faster R-CNN [28] and Mask R-CNN [29] is presented in Figure 1, where it can be divided into three main components. (1) Feature extraction: we employed ResNet [30] and FPN [31] as feature extractors for Faster R-CNN and Mask R-CNN. The feature extractor sends five feature maps of various sizes to the following network after feature extraction. (2) Region proposal networks (RPNs): five identical RPNs obtained five feature maps, which were then utilized to generate region proposals. Specifically, RPN generates anchors of different sizes to obtain the specified number of region proposal feature maps. (3) Region CNN (R-CNN): the R-CNN component of Faster R-CNN unifies the feature maps of the region proposal into the same size and then feeds them all into the fully connected layer for classification and regression. For Mask R-CNN, a full convolution branch is added to implement the instance segmentation task.

The two main differences between Mask R-CNN and Faster R-CNN are as follows. (1) The addition of the mask component (fully convolutional layer) to R-CNN enables the task of instance segmentation (Figure 1). (2) For the problem of misalignment between the feature map and the RoI on the original image, RoIAlign, which is an improved version of RoIPooling, is proposed (Figure 2).

The loss function of Faster R-CNN can be divided into two parts: R-CNN loss [32] and RPN loss [28], which is shown as Equations (1) and (2). For training, a joint training strategy will be used for both parts. Training Mask R-CNN only requires adding a loss of the mask component for training fully convolution layer, with the same joint training strategy as Faster R-CNN.
(1)$$L\left(p,u,t^{u},v\right)=L_{cls}\left(p,u\right)+\lambda^{\prime }L_{reg}\left(t^{u},v\right)$$
(2)$$L(\{p_{i}\},\{t_{i}\})=\frac{1}{N_{cls}}\sum_{i}L_{cls}(p_{i},p_{i}^{*})+\lambda \frac{1}{N_{reg}}\sum_{i}p_{i}^{*}L_{reg}(t_{i},t_{i}^{*})$$

The dataset contains 148 images of pavement cracks, in which 90% are training data and 10% are testing data. The images in the dataset are taken by smartphones, including single cracks, deep cracks, cracks with sunlight interference, bending cracks, etc. We used the random flip method for data augmentation, including horizontal and vertical flipping, with the probability set to 0.5, and as such our dataset was expanded by 50%. As shown in Figure 3, Faster R-CNN selects rectangles to label cracks in the image, and Mask R-CNN uses polygons to depict cracks in the image.

We use a backbone based on ImageNet pre-training and perform finetuning strategy to train our network. The input image size was set to 800 × 500 to ensure the model training speed (also limited by GPU memory), although our approach does not require specifying the size of the input image. We set five different learning rates for training both networks: 0.02, 0.01, 0.005, 0.0025 and 0.0002. We employed an SGD optimizer for training. In addition, methods, such as batch normalization and warm-up, are used in the training process to enhance the effect. We used PyTorch 1.8 and CUDA 11.1 for training. The task was performed on the Ubuntu 20.04.3 working platform and trained on a single NVIDIA GTX 1080 Ti GPU.

## 3. Analysis

We conducted a comparison using the popular method YOLO. The P-R curves of bounding-box of the three methods are shown in Figure 4, and it is obvious that Faster R-CNN has an advantage in detection, while YOLO v3 is hardly competent for our task. In addition, YOLOv3 requires a longer training time. It seems that using YOLO v3 for the specific task of pavement crack detection and with only a small amount of data do not yield the desired results.

Joint training is a common strategy in deep learning that can speed up training and often obtain good results. However, for Faster R-CNN and Mask R-CNN models with upstream and downstream relationships, it is not known how much the results will be affected by simply summing their losses and using joint training. The following experiment compares the detection of cracks by Faster R-CNN and Mask R-CNN after setting different learning rates. Five learning rates were set: 0.02, 0.01, 0.005, 0.0025, and 0.0002.

The learning rates of Faster R-CNN and Mask R-CNN are investigated and compared, which are presented in Figure 5 and Figure 6. The comparison of acc and loss plots based on different learning rates of Faster R-CNN and Mask R-CNN are discussed in this paper.

From the figures, it can be seen that the acc and loss of both algorithms do not change much after adjusting the learning rate. The acc values of both Faster R-CNN and Mask R-CNN are stable at approximately 98, the loss of Faster R-CNN is stable at approximately 0.25, and the loss of Mask R-CNN is stable at approximately 0.4. However, when the learning rate is 0.0002, the results are poorer, as seen by the comparison of the previous groups of detection results, but the acc and loss plots have more desirable results, which shows that the model will be overfitted when the learning rate is adjusted to 0.0002.

In Figure 5, different learning rates of Faster R-CNN and Mask R-CNN are presented where Figure 5a shows the acc comparison of five learning rates for Faster R-CNN; Figure 5b shows the acc comparison of five learning rates for Mask R-CNN; Figure 5c shows the loss comparison of five learning rates for Faster R-CNN; and Figure 5d shows the loss comparison of five learning rates for Mask R-CNN.

The comparison of accuracy vs. loss plots for the same learning rate of Faster R-CNN and Mask R-CNN are presented. Figure 6 shows the comparison of acc and loss plots for the same learning rate of five groups of Faster R-CNN and Mask R-CNN. As seen from the figure: the learning rate is 0.02, and 0.01 acc are shown as follows: the initial stage Mask R-CNN is higher, with the increase of iter, slightly decreases, and later Faster R-CNN is significantly higher; the learning rate is 0.005 when the difference between the two is not significant; the learning rate is 0.0025 when the initial Mask R-CNN is higher, and, later, the difference between the two is not significant. The loss of Faster R-CNN is always lower than that of Mask R-CNN, and the loss of Faster R-CNN is stable, at approximately 0.25, and the loss of Mask R-CNN is stable, at approximately 0.4. The loss of Faster R-CNN is always lower than that of Mask R-CNN. Since the loss function of Mask R-CNN increases, $L_{\mathrm{mask}}$, the loss value of Mask R-CNN will be larger.

In Faster R-CNN, each ROI has two outputs: one output is the classification result, which is the label of the prediction frame, and the other output is the regression result, which is the coordinates of the prediction frame. However, Mask R-CNN adds a third output, the object mask, which means that a mask is output for each ROI, and the branch is implemented through the FCN network. The loss function of Mask R-CNN is 1.2 and consists of three parts: $L_{cls}$, $L_{box}$ and $L_{\mathrm{mask}}$, where $L_{cls}$ and $L_{box}$ are consistent with the classification and regression losses defined in Faster R-CNN. In Mask R-CNN, for the newly added mask branches, the output dimension of each ROI is K × m × m, where m × m denotes the size of the mask and K denotes K categories, so a total of K binary masks are generated here. After obtaining the predicted mask, the sigmoid function value is found for each pixel point value of the mask, and the obtained result is used as one of the inputs to the $L_{\mathrm{mask}}$ (cross-entropy loss function). It should be noted that only the positive sample ROI is used to calculate $L_{\mathrm{mask}}$, and the definition of a positive sample is the same as the target detection, which is defined as a positive sample with an IOU greater than 0.5. In fact, $L_{\mathrm{mask}}$ is very similar to $L_{cls}$, except that the former is calculated based on pixel points and the latter is calculated based on images, so similar to $L_{cls}$, although K masks are obtained here, only the mask corresponding to the ground truth is valid in the calculation of the cross-entropy loss function.

In Figure 6, this set of images shows the acc and loss comparison between Faster R-CNN and Mask R-CNN with the same learning rate, where Figure 6a shows the acc and loss comparison between Faster R-CNN and Mask R-CNN when the learning rate is 0.02; Figure 6b shows the acc and loss comparison between Faster R-CNN and Mask R-CNN when the learning rate is 0.01; Figure 6c shows the acc and loss comparison between Faster R-CNN and Mask R-CNN when the learning rate is 0.005; Figure 6d shows the acc and loss comparison between Faster R-CNN and Mask R-CNN when the learning rate is 0.0025; and Figure 6e shows the acc and loss comparison between Faster R-CNN and Mask R-CNN when the learning rate is 0.0002.

## 4. Results and Discussion

Four different sets of results were selected for comparison: single crack vs. bifurcated crack comparison, crack with or without sunlight interference comparison, deep and shallow crack comparison, and straight and curved crack comparison.

### 4.1. Single Crack and Bifurcation Crack

Figure 7a shows the detection of bifurcation cracks by Faster R-CNN with different learning rates; Figure 7b shows the detection of single cracks by Faster R-CNN with different learning rates. As seen from the detection results, Faster R-CNN has little effect on a single crack after adjusting the learning rate, and the detection results are all better with a higher score-thr; for the bifurcated crack, Faster R-CNN is more sensitive to the change in learning rate and can only detect part of the crack, in which the detection results of the transverse crack are better than those of the vertical crack. The direction of the bifurcation crack is different, and it can be seen from the detection results that transverse cracks can be detected, while vertical cracks, branch cracks and small cracks cannot be detected completely. The detection results in this group of comparison tests when the learning rate is set to 0.005 are better; all cracks can be detected, and the score-thr is also higher. Figure 7c shows the detection of bifurcation cracks by Mask R-CNN with different learning rates; Figure 7d shows the detection of single cracks by Mask R-CNN with different learning rates. As seen from the detection results, Mask R-CNN is also more sensitive to single cracks after adjusting the learning rate. The learning rate is set to 0.02, 0.01, 0.005, and 0.0025, most of the cracks can be detected, and only a small local area is missed. When the learning rate is set to 0.0002, cracks are completely undetectable, and the model appears to be overfitted at this point. For bifurcation cracks, the detection of the five learning rates under this experiment is not very good, in which the detection results of vertical cracks are worse than those of transversal cracks, the detection effect becomes significantly worse when the learning rate is gradually reduced, the number of cracks that can be detected decreases, only sporadic segments of cracks can be detected, and cracks cannot be detected at all when the learning rate is reduced to 0.0002. From the results, it can be seen that, as in the case of Faster R-CNN, the detection results of single transverse cracks are all better; the detection effect of transverse cracks in bifurcated cracks is better than that of other directions. Overall, both algorithms are ideal for the detection of single cracks; complex bifurcated cracks require more data sets for training, and the training set used in this experiment has more transverse single cracks.

In Figure 7 Faster R-CNN and Mask R-CNN are compared, and the detection results of four groups with different learning rates are presented, where Figure 7a for the detection of bifurcated cracks by Faster R-CNN; Figure 7b for the detection of single cracks by Faster R-CNN; Figure 7c for the detection of bifurcated cracks by Mask R-CNN; and Figure 7d for the detection of single cracks by Mask R-CNN, the learning rates of each group of images from top to bottom are 0.02, 0.01, 0.005, 0.0025, and 0.0002.

### 4.2. Cracks with or without Sunlight Interference

We selected a set of comparison pictures, one of which is a road crack with sunlight interference, and the other is a picture of a normal road crack. Figure 8a shows Faster R-CNN for detecting cracks with sunlight interference; and Figure 8b shows Faster R-CNN for detecting cracks without sunlight interference. From the results, it can be seen that Faster R-CNN has better results for different learning rates for detecting well-lit cracks, and some cracks are not detected when the learning rate is set to 0.0002 for detecting poorly lit cracks. The comparison between a single crack and a bifurcated crack shows that the detection result of a transverse single crack is better, and a transverse single crack is chosen for this comparison, so the influence of the crack direction on the detection result is excluded in this experiment, and cracks with or without sunlight interference are not too sensitive to the learning rate. Figure 8c shows the crack with sunlight interference detection by Mask R-CNN; Figure 8d shows the crack without sunlight interference detection by Mask R-CNN. As seen from the results, when Mask R-CNN is used to detect well-lit cracks, the learning rate is set to 0.02, and part of the crack on the right is not detected; when the learning rate is reduced to 0.01, 0.005, and 0.0025, the whole crack can be detected; after continuing to reduce the learning rate to 0.0002, the model shows an overfitting phenomenon, and the detection effect is poor, with large segments of cracks not detected, and the detection score-thr is also reduced. When used to detect cracks without sunlight interference, the cracks can be detected completely at learning rates of 0.02, 0.01, and 0.005, and a small portion of cracks are not detected at the other two learning rates, where the score-thr of the model detecting cracks is very low at a learning rate of 0.0002. In general, both cracks with and without sunlight interference comparisons selected for this set of experiments had relatively good detection results.

In Figure 8 Faster R-CNN and Mask R-CNN are compared, and the detection results of four groups with different learning rates are presented, where Figure 8a for Faster R-CNN to detect cracks with sunlight interference; Figure 8b for Faster R-CNN to detect cracks without sunlight interference; Figure 8c for Mask R-CNN to detect cracks with sunlight interference; and Figure 8d for Mask R-CNN to detect cracks without sunlight interference, the learning rates of each group of images from top to bottom are 0.02, 0.01, 0.005, 0.0025, and 0.0002.

### 4.3. Deep and Shallow Cracks

Figure 9a shows the detection of deep cracks by Faster R-CNN and Figure 9b shows the detection of shallow cracks by Faster R-CNN. From the results, it can be seen that Faster R-CNN is used to detect deep and shallow cracks, different learning rates are set to detect the results more satisfactorily, and the cracks can be detected completely. The crack structure selected for this group of comparison experiments is relatively simple and belongs to a single crack in the transverse direction. Only learning rates of 0.02 and 0.0002 in the two groups of comparisons detect a small section of crack on the left side when a shallow crack is detected. It can also be seen from the comparison of the two previous groups that the model trained from this experimental data set is better for simple crack detection, and thus, for detecting deep and shallow cracks, Faster R-CNN is not sensitive to the learning rate. Figure 9c shows the detection of deep cracks by Mask R-CNN; Figure 9d shows the detection of shallow cracks by Mask R-CNN. As seen from the results, Mask R-CNN for detecting deep cracks is more sensitive to the change in learning rate than Faster R-CNN. When the learning rate is set to 0.02, the most complete cracks are detected, and as the learning rate is tuned down, small sections of cracks are missed. When the learning rate is 0.0002, the detection effect is the worst, and cracks cannot be detected. When used to detect shallow cracks, the learning rate of 0.005 has the best effect, and the detected cracks are the most complete. The other learning rates also have the situation of missed detection, especially when the learning rate is set to 0.0002, the model exhibits an overfitting phenomenon, and cracks cannot be detected. In the comparison of deep and shallow cracks in this experiment, Faster R-CNN has better detection results than Mask R-CNN. For deep and shallow cracks, Faster R-CNN can basically detect them, while Mask R-CNN is more sensitive to the change in learning rate. There are many missed detections after setting an inappropriate learning rate, and the detection results become worse.

In Figure 9 Faster R-CNN and Mask R-CNN are compared, and the detection results of four groups with different learning rates are presented, where Figure 9a for Faster R-CNN to detect deep cracks; Figure 9b for Faster R-CNN to detect shallow cracks; Figure 9c for Mask R-CNN to detect deep cracks; and Figure 9d for Mask R-CNN to detect shallow cracks, the learning rates of each group of images from top to bottom are 0.02, 0.01, 0.005, 0.0025 and 0.0002.

### 4.4. Straight and Bending Cracks

Figure 10a shows the detection of straight cracks by Faster R-CNN; Figure 10b shows the detection of curved cracks by Faster R-CNN. As seen from the results, Faster R-CNN adjusts the learning rate and has little effect on horizontal straightaway cracks; the detection results are better, and the score-thr is higher. When detecting bending cracks, Faster R-CNN is more sensitive to the learning rate change. When the learning rate is set to 0.02, all the cracks can be detected, and when the learning rate is reduced, each image will be partially missed. Figure 10c shows the detection of straight cracks by Mask R-CNN and Figure 10d shows the detection of bending cracks by Mask R-CNN. As seen from the results, Mask R-CNN is used to detect transverse straight cracks with better results than bending cracks, most of the transverse cracks—which have a simpler crack form—can be detected, and only a small portion of them are missed. When the learning rate is set to 0.0002, the detection effect is the worst, and the score-thr of the detection result is very low; even cracks cannot be detected. The other four learning rates set in this experiment have similar detection situations, and there will be small sections of missed detection, where the learning rate is 0.01, the detection result is the best, and the detected cracks are the most complete. The five learning rates set in this experiment are not ideal for detecting bending cracks, and cracks cannot be detected completely, as shown in the figure. Faster R-CNN and Mask R-CNN both have better detection results for transverse and simple cracks, and more crack data may be needed to train the model for complex cracks.

In Figure 10, Faster R-CNN and Mask R-CNN are compared, and the detection results of four groups with different learning rates are presented, where Figure 10a for Faster R-CNN to detect laterally biased cracks; Figure 10b for Faster R-CNN to detect curved cracks; Figure 10c for Mask R-CNN to detect laterally biased cracks; and Figure 10d for Mask R-CNN to detect curved cracks, the learning rates of each group of images from top to bottom are 0.02, 0.01, 0.005, 0.0025, and 0.0002.

### 4.5. Effectiveness of Detection for Other Dataset

We tested the CRACK500 dataset using the completed training model, and some of the results are shown in Figure 11 and Figure 12. Faster R-CNN still performs well on different datasets with strong generalization performance. Mask R-CNN performs slightly worse, but is still valid. It should be noted that migration-learning based finetuning methods are very efficient. It is reasonable to believe that we will obtain even better results if we train a small amount for the CRACK500 dataset.

## 5. Conclusions

With the rapid increase in road mileage, the traditional road crack monitoring method has difficulty meeting the demand, and intelligent monitoring technology is becoming increasingly urgent. In this study, deep learning methods are investigated to detect road cracks with the comparison of Faster R-CNN and Mask R-CNN. The following points in this study can be concluded.

(i) For the same crack image with the same set of data training, the detection bounding-boxes of Faster R-CNN are more complete than those of Mask R-CNN, and in this study, in which the joint training strategy leads to difficulties in its bounding box regression, the score-thr of Faster R-CNN is higher. At the same time, we see that this type of training is effective. Both models are able to detect cracks better with only 130+ images trained, and the use of SGD optimizer makes the models insensitive to the learning rate setting. However, the detection of more complex cracks is more sensitive to the learning rate, and the worst detection result is obtained when the learning rate is 0.0002.

(ii) Our method works better than YOLOv3, as both models are ideal for the detection of single cracks, and both algorithms have ideal detection results for detecting bright and dark cracks. Additionally, both algorithms can detect straight cracks well. For the detection of deep and shallow cracks, Faster R-CNN performs well in this dataset.

(iii) Both Faster R-CNN and Mask R-CNN require more datasets to train for complex cracks, and the dataset in this experiment has more transverse single cracks and less complex cracks.

(iv) The acc and loss of Faster R-CNN do not differ much under the five learning rates set in this experiment, and the case of Mask R-CNN is the same as that of Faster R-CNN.

(v) At the same learning rate, the acc of Faster R-CNN is higher than that of Mask R-CNN when the learning rate is more than 0.005, and the learning rate of Mask R-CNN is higher than that of Faster R-CNN when the learning rate is less than 0.005; the loss value of Faster R-CNN is smaller than that of Mask R-CNN.

For further work, how to use a small amount of labeled data more effectively to achieve better results will be our research goal. More suitable pre-trained models, active learning, data augmentation, etc., may be useful for the goal. Considering that deep learning methods are constantly evolving and the algorithms are improving, we will investigate more perfect algorithms or improve existing algorithms to obtain more advanced detection results in the future.

## Figures and Tables

**Figure 1 sensors-22-01215-f001:**
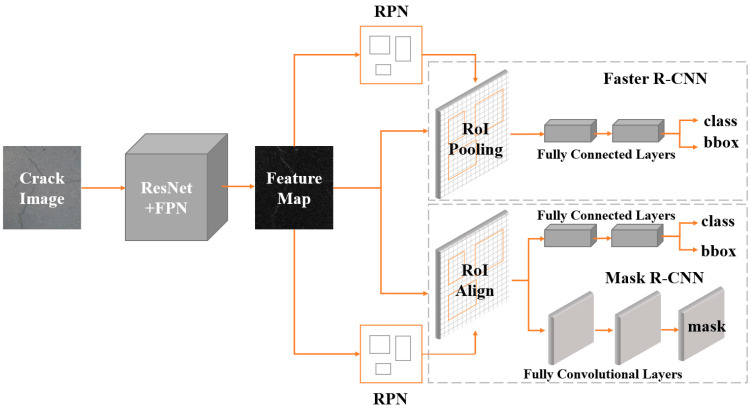
Pipeline of Faster R-CNN and Mask R-CNN: RPN is trained separately. In addition, the mask branch and bbox branch use different RoI feature extractors, and the mask branch obtains better results through a larger feature map.

**Figure 2 sensors-22-01215-f002:**
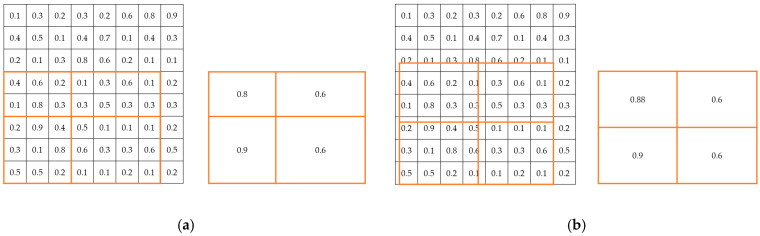
Visualization diagram of RoI Pooling and RoI Align: Assume that the output size is (2, 2). RoIAlign uses a bilinear interpolation method to improve accuracy. (**a**) RoI Pooling. (**b**) RoI Align.

**Figure 3 sensors-22-01215-f003:**
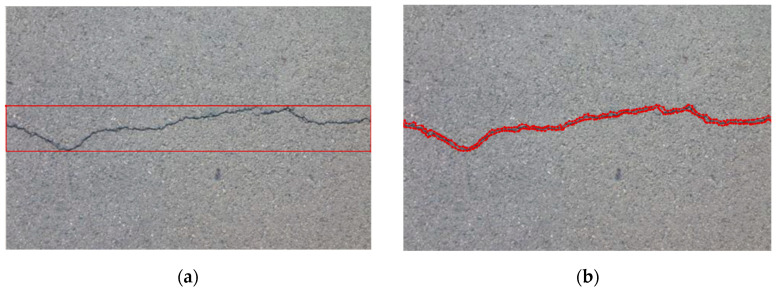
Labeling of cracks. (**a**) Labeling of Faster R-CNN; (**b**) Labeling of Mask R-CNN.

**Figure 4 sensors-22-01215-f004:**
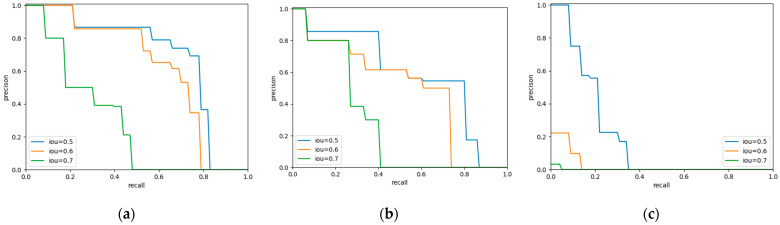
Precision-recall curve of bounding-boxes in different models. (**a**) Faster R-CNN. (**b**) Mask R-CNN. (**c**) YOLO v3.

**Figure 5 sensors-22-01215-f005:**
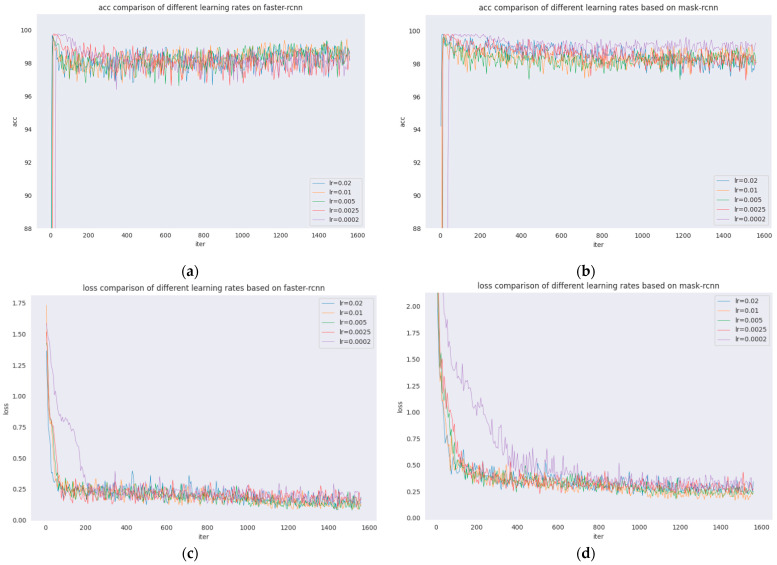
Various learning rate analyses. (**a**) Accuracy of various learning rates of Faster R-CNN. (**b**) Accuracy of various learning rates of Mask R-CNN. (**c**) Loss of various learning rates of Faster R-CNN. (**d**) Loss of various learning rates of Mask R-CNN.

**Figure 6 sensors-22-01215-f006:**
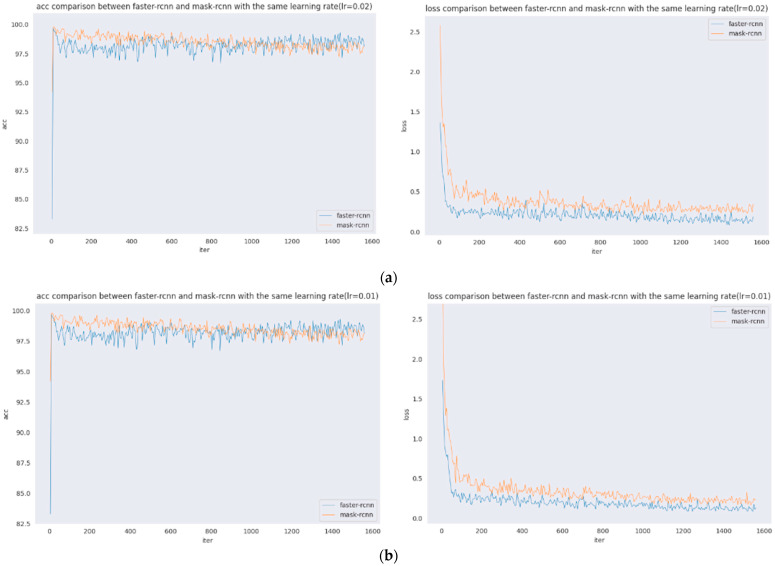
Accuracy and loss comparison with the same learning rate. (**a**) Accuracy and loss comparison (lr = 0.02). (**b**) Accuracy and loss comparison (lr = 0.01). (**c**) Accuracy and loss comparison (lr = 0.005). (**d**) Accuracy and loss comparison (lr = 0.0025). (**e**) Accuracy and loss comparison (lr = 0.0002).

**Figure 7 sensors-22-01215-f007:**
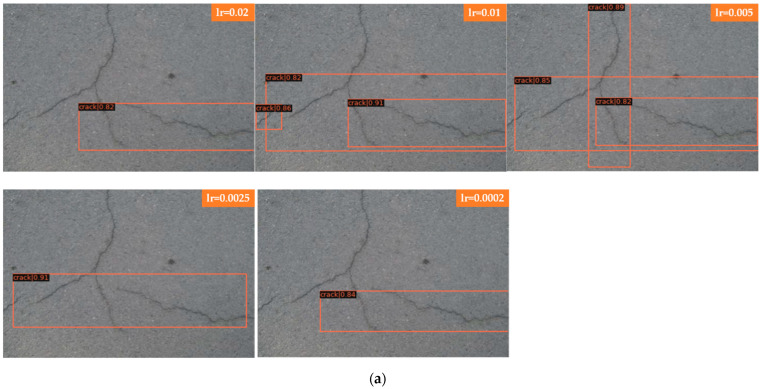
Comparison of bifurcations and single cracks. (**a**) Faster R-CNN for bifurcation crack. (**b**) Faster R-CNN for single crack. (**c**) Mask R-CNN for bifurcation crack. (**d**) Mask R-CNN for single crack.

**Figure 8 sensors-22-01215-f008:**
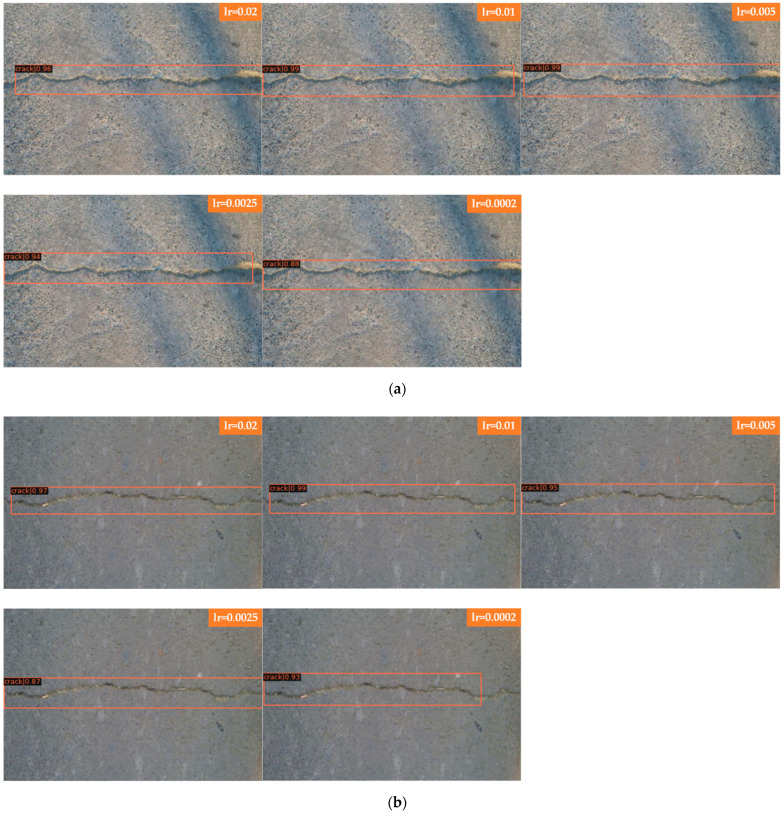
Comparison of road crack detection results with and without sunlight interference. (**a**) Faster R-CNN for cracks with sunlight interference. (**b**) Faster R-CNN for cracks without sunlight interference. (**c**) Mask R-CNN for cracks with sunlight interference. (**d**) Mask R-CNN for cracks without sunlight interference.

**Figure 9 sensors-22-01215-f009:**
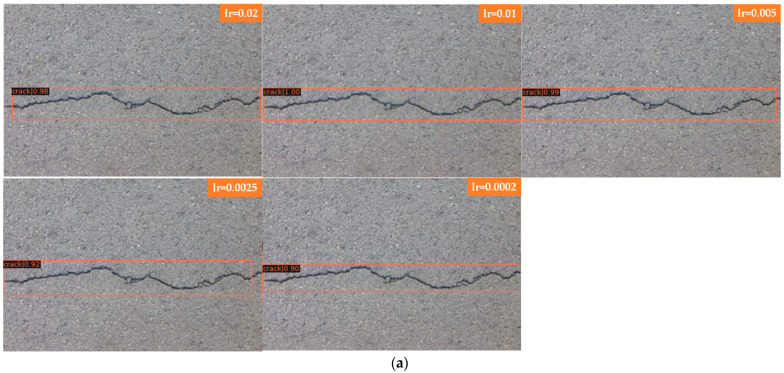
Comparison of deep and shallow cracks. (**a**) Faster R-CNN for deep cracks. (**b**) Faster R-CNN for shallow cracks. (**c**) Mask R-CNN for deep cracks. (**d**) Mask R-CNN for shallow cracks.

**Figure 10 sensors-22-01215-f010:**
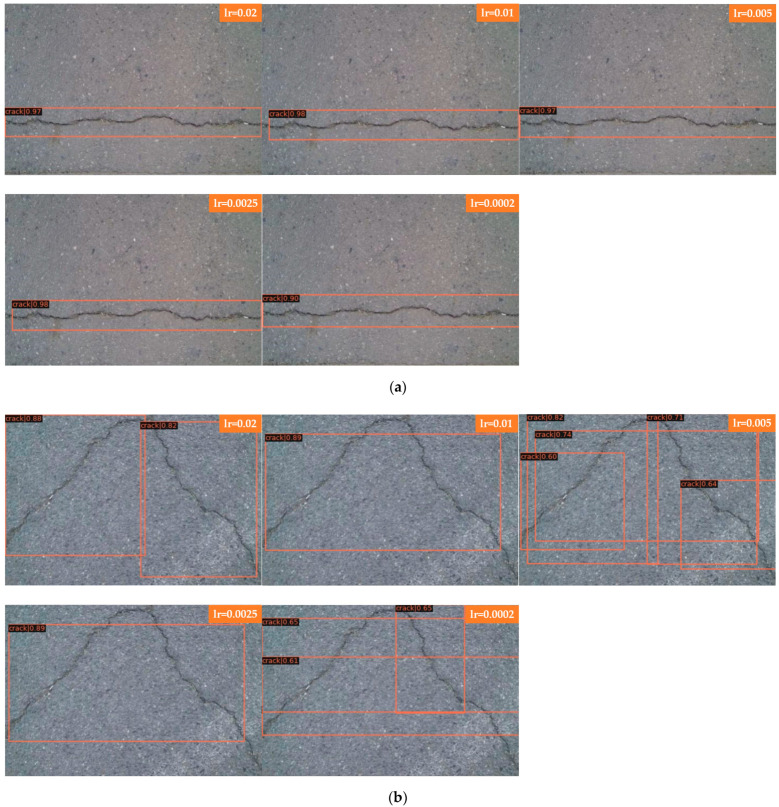
Comparison of straight and bending cracks. (**a**) Faster R-CNN for straight cracks. (**b**) Faster R-CNN for bending cracks. (**c**) Mask R-CNN for straight cracks. (**d**) Mask R-CNN for bending cracks.

**Figure 11 sensors-22-01215-f011:**
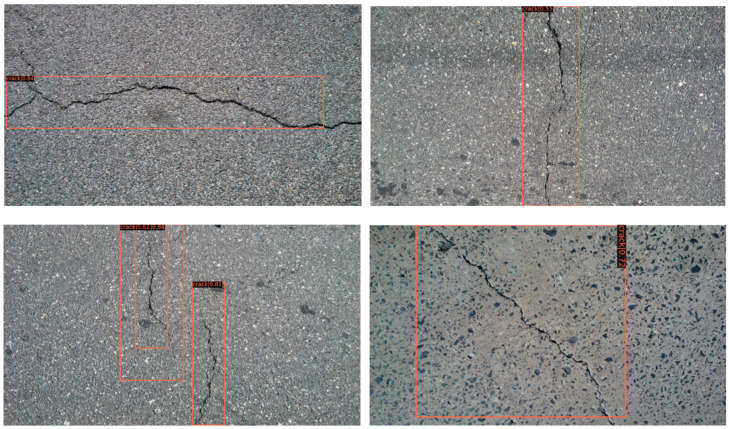
Test results using Faster R-CNN.

**Figure 12 sensors-22-01215-f012:**
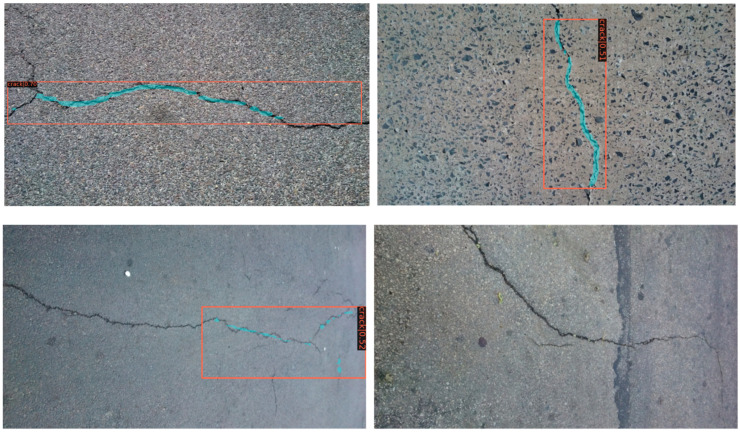
Test results using Mask R-CNN.

## Data Availability

The data presented in this study are available on request from the corresponding author.

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
