# Peer review of "Crack Detection and Comparison Study Based on Faster R-CNN and Mask R-CNN"

_sensors, 2022, doi:10.3390/s22031215_

Round 1

Reviewer 1 Report

My review comments:

This paper deals with image-based crack detection by means of two types of convolutional neural networks, namely Faster R-CNN and Mask R-CNN. The experimental results show that the Faster R-CNN model is more reliable than the Mask R-CNN model. Overall, the topic of this study is interesting. However, some technical issues should be well clarified before it is accepted for publication. 

The detailed comments  are given as follows. 

  1. The title is rather misleading as it does not express what the manuscript is about. The term intelligent is not adequate in this context.
  2. The contribution and innovation of the manuscript should be clarified more clearly in abstract and introduction. 
  3. The literature review is not discussed comprehensively. Please do not only present the previous related papers, but also point out their research deficiencies. And then highlight the innovation in your research. 
  4. How did the authors implement the numerical experiments?  There is no information about software implementation of the models and experiments.
  5. It is advised to perform a cross-data set study to show the ability of the trained networks, using for example SDNET2018 data set. 
  6. To estimate the generalization, you should also apply other metrics such as precision, recall and the F1 score.
  7. More future research should be included in the conclusion part.

Author Response

Dear Reviewer,

Thank you very much for your comments. Your comments have been answered as follows.

  1. The title is rather misleading as it does not express what the manuscript is about. The term intelligent is not adequate in this context.

Re: Thanks a lot. We have changed the title in the revised version as “Crack detection and comparison study based on Faster -R-CNN and Mask -R-CNN”.

  1. The contribution and innovation of the manuscript should be clarified more clearly in abstract and introduction. 

Re: Thanks a lot. We have revised the abstract and introduction in the revised version, thank you for your suggestions.

  1. The literature review is not discussed comprehensively. Please do not only present the previous related papers, but also point out their research deficiencies. And then highlight the innovation in your research. 

Re: Thanks a lot. The deficiencies of previous research are that the crack detection research considering the characteristics of road images is still insufficient which has been added in the revision. The literature review has been revised. Thank you very much.

  1. How did the authors implement the numerical experiments?  There is no information about software implementation of the models and experiments.

Re: Thanks a lot. We have clarified the software, environment, and platform which we used in the revision. We used PyTorch 1.8 and CUDA 11.1 for training. The task was done at Ubuntu 20.04.3 working platform.

  1. It is advised to perform a cross-data set study to show the ability of the trained networks, using for example SDNET2018 data set. 

Re: Thank you very much. We tested the CRACK500 dataset and the results proved the generalization performance of our model.

  1. To estimate the generalization, you should also apply other metrics such as precision, recall and the F1 score.

Re: Thanks a lot. The Precision-Recall curve and a brief analysis are added which are helpful to clarify the detection effect of the model. Thank you very much.

  1. More future research should be included in the conclusion part.

Re: Thanks a lot. We have rewritten the future work in the conclusion part as “For further work, how to use a small amount of labeled data more effectively to achieve better results will be our research goal. More suitable pre-trained models, active learning, data augmentation, etc. may be useful for the goal. Considering that deep learning methods are constantly evolving and the algorithms are improving, we will investigate more perfect algorithms or improve existing algorithms to obtain more advanced detection results in the future.”

Thanks a lot and have a nice day.

Best regards,

Xiangyang Xu

Reviewer 2 Report

This paper used Fast R-CNN and Mask R-CNN for pavement crack recognition. Generally, the background, methodology and results are clearly presented. Have several comments as below

  1. Both fast R-CNN and mask R-CNN have been used for pavement crack recognition. Need to highlight the contribution of this paper.
  2. Figures 3-6 can be simplified. Can you explain why the detection effect of mask R-CNN is poorer?
  3. English expression must be improved.
  4. The dataset contains 148 images in which 90% are training data and 10% are testing data. Does it too small for training and testing?
  5. In section 3.1, how the cracks are labeled in the training set?
  6. In section 3.2, what are the standard for distinguishing between good and low light?
  7. In section 4, the loss comparison seems irrelevant to the topic of the paper.

Author Response

Dear Reviewer,

Thank you very much for your comments. Your comments have been answered as follows.

  1. Both fast R-CNN and mask R-CNN have been used for pavement crack recognition. Need to highlight the contribution of this paper.

Thank you very much. We have added this part in the revised version. We compared the two methods, both of which are identical in the part of generating bounding-box, and the test results showed that their identical parts did not achieve the same results, for which we analyzed and explained. In addition, we adopted some new training methods that allow us to achieve better results with only a small amount of data.

  1. Figures 3-6 can be simplified. Can you explain why the detection effect of mask R-CNN is poorer?

Thanks a lot. Figure 3-6 shows the detection results obtained with different learning rates, we add the annotation, Mask R-CNN is poorer because we use joint training, which makes the regression of bounding-box slightly poor, we have explained in the revised manuscript.

  1. English expression must be improved.

Thank you for pointing out the problem, we improved our English expressions and made revisions.

  1. The dataset contains 148 images in which 90% are training data and 10% are testing data. Does it too small for training and testing?

Thanks a lot. Because the roads are generally in better condition now, the amount of crack data is on the low side. We are hoping to get good results with a small amount of data.

  1. In section 3.1, how the cracks are labeled in the training set?

Thanks a lot. We have added a description of data annotation in the revision as “As shown in the Figure 3, Faster R-CNN selects rectangles to label cracks in the image, and Mask R-CNN uses polygons to depict cracks in the image.”

  1. In section 3.2, what are the standard for distinguishing between good and low light?

Thank you for pointing out the problem. We meant with or without sunlight interference. The former shows sunlight influence and there are shadows of disturbing objects in the image, while the latter doesn’t and there is no shadow of disturbing objects in the image. We have made a correction in the revised version.

  1. In section 4, the loss comparison seems irrelevant to the topic of the paper.

Thanks again for your comments, it was mentioned earlier that Mask R-CNN and Faster R-CNN showed different results with the same sub model, which we think is related to the joint training strategy. Observing the variation of Loss helps to demonstrate this phenomenon, and we have corrected it in the revised version.

Thanks a lot and have a nice day.

Best regards,

Xiangyang Xu

Reviewer 3 Report

The manuscript applies existing AI/ML techniques to crack detection. Experimentations are done well and the manuscript is also written well however i have the following suggestions. 

  1. Recently researchers used some techniques based on stats of boxes to improve the algorithm proposed. pls check https://www.mdpi.com/2504-2289/6/1/9 What will be the effect of such improvement in the proposed algorithm.
  2. What if the author uses Deep CNN in place of traditional algo used in the paper for eg https://www.mdpi.com/1999-5903/13/12/307 Add one or two lines of comparison.
  3. why authors have not used YOLO which is proved to be more efficient for eg ref https://www.mdpi.com/2079-9292/10/24/3079
  4. What about using XAI method like GRAD-CAM to visually explain findings of the algorithm for eg ref https://arxiv.org/abs/1610.02391
  5. ROC curves to be plotted and explained.
  6. Authors should mention about validation methods used ref can be found at https://www.sciencedirect.com/science/article/pii/S0164121221001473
  7. Data generated is less for ML applications. Authors should comment on augmentation methods for future work. Ref https://journalofbigdata.springeropen.com/articles/10.1186/s40537-019-0197-0
  8. If there are multiple defects proper segmentation of the image will help. Author should write about the same https://link.springer.com/chapter/10.1007/978-981-13-0761-4_105 or https://www.sciencedirect.com/science/article/pii/S1877050915028574
  9. Lot of grammatical errors to be improved.
  10. Authors need to explain confusion matrics parameters clearly and significance of them. Ref https://arxiv.org/ftp/arxiv/papers/2001/2001.09636.pdf

Author Response

Dear Reviewer,

Thank you very much for your comments. Your comments have been answered as follows.

  1. Recently researchers used some techniques based on stats of boxes to improve the algorithm proposed. pls check https://www.mdpi.com/2504-2289/6/1/9 What will be the effect of such improvement in the proposed algorithm.

Thanks a lot. We have added the citation in our revision, and some of the problems it mentions we encountered, such as the anchor ratio problem.

  1. What if the author uses Deep CNN in place of traditional algo used in the paper for eg https://www.mdpi.com/1999-5903/13/12/307 Add one or two lines of comparison.

Thanks a lot. We include the experimental results of YOLO v3 in the revised version, and the results show that YOLO v3 is hardly competent for our task and it converges more slowly.

  1. why authors have not used YOLO which is proved to be more efficient for eg ref https://www.mdpi.com/2079-9292/10/24/3079.

Thanks a lot. Same as question 2. We include the experimental results of YOLO v3 in the revised version, and the results show that YOLO v3 is hardly competent for our task and it converges more slowly.

  1. What about using XAI method like GRAD-CAM to visually explain findings of the algorithm for eg ref https://arxiv.org/abs/1610.02391.

Thanks a lot. GRAD-CAM is a good visualization method but is usually used in image classification tasks. Anchor-based method (such as Faster R-CNN) generates region proposals by anchor and determines whether it is a positive or negative sample, and finally obtains the final bounding boxes by some methods such as NMS. We are not classifying in the bounding box region, so it may cause ambiguity if CAM is used.

  1. ROC curves to be plotted and explained.

Thanks a lot. We have added Precision-Recall curves to the revised version and performed a brief analysis.

  1. Authors should mention about validation methods used ref can be found at https://www.sciencedirect.com/science/article/pii/S0164121221001473.

Thanks a lot. We have added the citation in the revision and appreciate your suggestions.

  1. Data generated is less for ML applications. Authors should comment on augmentation methods for future work. Ref https://journalofbigdata.springeropen.com/articles/10.1186/s40537-019-0197-0.

Thanks a lot. We used random flip for data augmentation and described it in the revision. It is really a useful approach for small dataset.

  1. If there are multiple defects proper segmentation of the image will help. Author should write about the same https://link.springer.com/chapter/10.1007/978-981-13-0761-4_105 or https://www.sciencedirect.com/science/article/pii/S1877050915028574.

Thanks a lot. We read the paper carefully, and this is indeed a good approach. However, we would like to minimize the manual operation and automate the detection, and in fact the anchor-based approach we used can be effective.

  1. Lot of grammatical errors to be improved.

Thanks a lot. We have checked the grammatical issues in the paper and fixed them, thank you for pointing them out.

  1. Authors need to explain confusion metrics parameters clearly and significance of them. Ref https://arxiv.org/ftp/arxiv/papers/2001/2001.09636.pdf.

Thanks a lot. Confusion metrics is a good tool. However, our task contains only one class, and we plotted Precision-Recall curves that should illustrate the detection effect.

Thanks a lot and have a nice day.

Best regards,

Xiangyang Xu

Round 2

Reviewer 2 Report

All comments are addressed

Reviewer 3 Report

well done.